# Impact of Newborn Screening on Clinical Presentation of Congenital Adrenal Hyperplasia

**DOI:** 10.3390/medicina57101035

**Published:** 2021-09-29

**Authors:** Rūta Navardauskaitė, Kornelija Banevičiūtė, Jurgita Songailienė, Kristina Grigalionienė, Darius Čereškevičius, Marius Šukys, Giedrė Mockevicienė, Marija Smirnova, Algirdas Utkus, Rasa Verkauskienė

**Affiliations:** 1Institute of Endocrinology, Medical Academy, Lithuanian University of Health Sciences, LT-50009 Kaunas, Lithuania; k.baneviciute@gmail.com (K.B.); rasa.verkauskiene@lsmuni.lt (R.V.); 2Medical Faculty, Institute of Biomedical Sciences, Vilnius University, LT-03101 Vilnius, Lithuania; jurgita.songailiene@santa.lt (J.S.); kristina.grigalioniene@santa.lt (K.G.); algirdas.utkus@mf.vu.lt (A.U.); 3Centre for Medical Genetics, Vilnius University Hospital Santaros Klinikos, LT-08661 Vilnius, Lithuania; marija.smirnova@santa.lt; 4Department of Genetics and Molecular Medicine, Medical Academy, Lithuanian University of Health Sciences, LT-50161 Kaunas, Lithuania; darius.cereskevicius@kaunoklinikos.lt (D.Č.); marius.sukys@lsmu.lt (M.Š.); 5Department of Endocrinology, Medical Academy, Lithuanian University of Health Sciences, LT-50161 Kaunas, Lithuania; giedre.mockeviciene@lsmu.lt

**Keywords:** newborn, screening, NBS, CAH, congenital adrenal hyperplasia

## Abstract

*Background and Objectives*: The main reason for Newborn screening (NBS) for congenital adrenal hyperplasia (CAH) is to prevent adrenal insufficiency that can lead to life-threatening conditions. On the other hand, screening programs are not always sensitive and effective enough to detect the disease. We aimed to evaluate impact of the national NBS on the clinical presentation of patients with CAH in Lithuania. *Materials and Methods*: A retrospective study was performed on data of 88 patients with CAH from 1989 to 2020. Patients with confirmed CAH were divided into two groups: (1) 75 patients diagnosed before NBS: 52 cases with salt-wasting (SW), 21 with simple virilising (SV) and two with non-classical (NC) form; (2) 13 patients diagnosed with NBS: 12 cases with SW and 1 case with SV form. For the evaluation of NBS effectiveness, data of only male infants with salt-wasting CAH were analysed (*n* = 36, 25 unscreened and nine screened). Data on gestational age, birth weight, weight, symptoms, and laboratory tests (serum potassium and sodium levels) on the day of diagnosis, were analysed. *Results*: A total of 158,486 neonates were screened for CAH from 2015 to 2020 in Lithuania and CAH was confirmed in 13 patients (12 SW, one–SV form), no false negative cases were found. The sensitivity and specificity of NBS program for classical CAH forms were 100%; however, positive predictive value was only 4%. There were no significant differences between unscreened and screened male infant groups in terms of age at diagnosis, serum potassium, and serum sodium levels. Significant differences were found in weight at diagnosis between the groups (−1.67 ± 1.12 SDS versus 0.046 ± 1.01 SDS of unscreened and screened patients respectively, *p* = 0.001). *Conclusions*: The sensitivity and specificity of NBS for CAH program were 100%, but positive predictive value—only 4%. Weight loss was significantly lower and the weight SDS at diagnosis was significantly higher in the group of screened patients.

## 1. Introduction

Congenital adrenal hyperplasia (CAH) is one of the life-threatening disorders in neonates. The term CAH refers to a group of autosomal recessive disorders that result from impaired steroidogenesis in the adrenal cortex. The estimated prevalence of CAH is 1:10,000 and annual incidence ranges from 1: 5000 to 1: 20,000 [1,2]. In 95% of cases, CAH is caused by mutation of the *CYP21A2* gene that encodes the enzyme 21-hydroxylase (21OH) [3,4]. Due to a deficiency of 21OH, the synthesis of cortisol and aldosterone is impaired. The clinical symptoms depend on specific *CYP21A2* mutations and range from mild to a life-threatening condition—adrenal insufficiency. As a result of impaired reverse negative feedback of cortisol to the pituitary gland, the secretion of adrenocorticotropic hormone (ACTH) is increased which stimulates testosterone hypersecretion, causing virilized genitalia in female newborns [4,5]. During the prenatal period, testosterone hypersecretion does not result in any visible signs for male infants and adrenal insufficiency occurs at 2 to 3 weeks of age with severe hyperkalemia, hyponatremia, and hypovolemic shock [6]. The different clinical presentations of CAH are classified as salt-wasting (SW), the most severe form, which may result in acute salt loss and a circulatory collapse during the first weeks of life, simple virilizing (SV)—the less severe form with prenatal virilization but without life-threatening salt loss, and the non-classical (NC) CAH—a mild form with slightly elevated androgen levels. More than 2/3 of CAH cases are severe SW forms of the disease. [7]. In order to prevent life-threatening adrenal crisis and to help make the appropriate sex assignments in affected female patients, newborn screening (NBS) programs for CAH have been introduced in many countries. Measurements of 17-hydroxyprogesterone (17OHP) levels have been used as a marker for the 21OH deficiency [7].

Since 2015, a national Newborn screening (NBS) program for CAH has been introduced in Lithuania. The aim of our study was to summarize the results of the past six years of NBS for CAH in Lithuania. Specifically, we wished to evaluate the efficiency of CAH screening and the prevalence/incidence of CAH in Lithuania.

## 2. Materials and Methods

### 2.1. Newborn Screening

Newborn screening (NBS) program for phenylketonuria, congenital hypothyroidism, congenital adrenal hyperplasia, and galactosemia in Lithuania is performed nationwide. NBS is managed by the Ministry of Health of Lithuania and funded by the National Health Insurance Fund under the Ministry of Health.

NBS tests are conducted by the single NBS laboratory at the Vilnius University Hospital Santaros Klinikos.

NBS for CAH in Lithuania started in 2015. Since then, more than 158,000 newborns have been screened in NBS laboratory, managing approximately 26,000 samples per year. Coverage of the NBS is 99.3–99.7% of newborns, despite the fact that NBS is not mandatory in Lithuania.

The dried blood spot (DBS) samples were collected from the infants on Whatman 903 filter paper on the 3–5th day of life at the time of the routine hospital newborn screening heel prick. DBS was requested to be delivered to the NBS laboratory within 72 h of collection by courier or regular post.

The level of 17-hydroxyprogesterone (17OHP) in the DBS was measured by fluorometric enzyme immunoassay (Neonatal 17OH Progesterone FEIA, Labsystems, Finland). As false-positive results are common with premature infants, a fixed cut-off level, based on gestational age (GA), was used to identify newborns at risk for CAH (Table 1). When elevated 17OHP was found, the first sample was reanalyzed. The newborns whose results were higher than cut-off were either recalled to repeat a test of 17OHP measurement or referred to a pediatric endocrinologist for clinical and laboratory evaluation (Table 1). Newborns with elevated 17OHP in the second DBS were also referred to a pediatric endocrinologist.

### 2.2. Biochemical Diagnosis of Salt-Wasting CAH

A case is considered true positive when laboratory tests are evaluated together with the clinical assessment of the child and data are consistent with the CAH diagnosis. Criteria of a positive screening test as true were elevated serum 17OHP >30 nmol/L (basal or during Synacten stimulation test) measured by radioimmunoassay (RIA) method. Hyperkalemia is defined as >6.9 mmol/L, hyponatraemia <136 mmol/L according to the reference range for neonatal age.

### 2.3. Molecular Diagnostics

All diagnosed patients and cases with uncertain diagnoses were genotyped for confirmation of the diagnosis. *CYP21A2* gene and *CYP21A1P* pseudogene copy number analysis was performed using quantitative multiplex ligation-dependent probe amplification (MLPA) with SALSA^®^ MLPA^®^ probemix P050-C1 CAH (MRC-Holland, Amsterdam, The Netherlands); reference sample—SD039-S02 Reference DNA (MRC-Holland). Detection of sequence changes of *CYP21A2* gene was performed using Sanger sequencing after selective long-range PCR with primers specific for *CYP21A2* and/or *CYP21A1P* [8].

### 2.4. Subjects

Data were collected retrospectively on 88 newborns diagnosed with CAH and followed up at the Lithuanian University of Health Sciences Kaunas Clinics Endocrinology department from 1989 to 2020.

Data collected for each patient included the birth date, sex, diagnosis (SW, SV, and NC), age at diagnosis and weight, symptoms, and laboratory tests (serum 17OHP, ACTH, potassium and sodium levels) on the day of admission to pediatric endocrinology department.

For the evaluation of NBS effectiveness, data of only male infants with SW CAH were analyzed whereas during the prenatal period, testosterone hypersecretion does not result in any visible signs for male infants and adrenal insufficiency occurs at 2 to 3 weeks of age with severe hyperkalemia, hyponatremia, and hypovolemic shock.

Patients with confirmed CAH were divided into 2 groups:(1)Before NBS: 75 patients; 52 cases with SW form, 21-with SV, and 2-with NC form.(2)After NBS: 13 patients; 12 cases with SW, and 1 case with SV form (Figure 1).

For the evaluation of NBS effectiveness, data of only male infants with SW CAH were analyzed.

### 2.5. Statistical Analyses

Statistical data analyses were performed using Statistical Package for Social Sciences (version 23, SPSS, Inc., Chicago, IL, USA). Birth weight was converted to standard deviation score according to gestational age while the weight at the day of diagnosis was converted to standard deviation score according to sex and age at the determination of CAH. Quantitative variables were described as a mean and standard deviation (M ± SD), nominal values were given in frequencies. The Mann—Whitney test was used for non-normal data distribution.

The differences were considered statistically significant when *p* < 0.05.

### 2.6. Bioethics

The study was approved by the local and national ethical committees (No. BE-2-29), and all patients and their parents or legal guardians gave their informed consent. The study was carried out in accordance with the 1964 Helsinki declaration and its later amendments.

## 3. Results

### 3.1. Newborn Screening for CAH

There were 158,987 live births registered in Lithuania between the 1st of January 2015 and 31st of December 2020. Of these, 158,486 (99.7%) were screened for CAH. The overall results of the screening for CAH from 2015 to 2020 are shown in Table 2.

The predictive value of a positive test for classical CAH forms was 0.04. The sensitivity was 1.0 as no false-negative cases were found, and the specificity was 0.998. Calculated incidence of classical CAH—1:12,190.

However, we cannot confirm the sensitivity of NBS for non-classical CAH form due to conditionally high cut-off of 17OHP serum concentrations in DBS, because the basal 17OHP concentration for NC CAH can be lower than the NBS cut-off (>6 nmol/L) [9].

In total, 320 infants required further testing: 204 newborns were recalled for a second sample and 116 newborns with abnormal 17-OHP levels on first DBS were referred for urgent medical evaluation, 2 newborns were referred for medical evaluation after 17OHP was repeatedly found elevated in the second sample. CAH was confirmed in 13 newborns. Gender as indicated on DBS cards: 9 males, 1 female, and 2 with ambiguous genitalia (later assigned as females). Twelve cases of the patients identified in the screening had the SW form of the disease, and 1 had SV with a mild clitoriomegaly but without any other clinical symptoms.

The median gestational age and birth weight of the true-positive CAH cases was 40 weeks (range: 38–41 weeks) and 3525 g (range: 3100–4088 g). On average, the screening DBS samples of confirmed CAH cases were collected on day 3 (median) (range: 2–4) and reached the screening laboratory by mail or courier on day 9 (range 3–19), and the 17OHP result was available on day 13 (range: 7–23 days). On receipt DBS, 17OHP results were available on 2nd or 4th day, as 17OHP assay with overnight incubation was performed twice per week. Three patients were suspected of having CAH on the first day of life: two newborns had ambiguous genitalia and one was the male sibling of a previously confirmed CAH patient. The median day of CAH diagnosis confirmation by pediatric endocrinologist was 15 (range: 8–32, mean 15.44 ± 7.79 days). The median screening value of 17-OHP in patients with SW CAH was 115 ng/mL (range: 29.19–651 ng/mL).

### 3.2. Clinical and Laboratory Assessment of Patients with SW CAH

Patients with CAH SW form before NBS (unscreened *n* = 52) and after NBS (screened *n* = 12) were compared. There were 34 male newborns (unscreened *n* = 25, screened *n* = 9) and 31 female newborns (unscreened *n* = 27, screened *n* = 3). The gestational age average of the unscreened cohort was 39.36 ± 1.25 weeks, that of screened cohort—39.75 ± 0.87 weeks, *p* = 0.329. The birth weight of unscreened patients was 0.11 ± 1.04 SDS and screened patients—0.077 ± 0.62 SDS, *p* = 0.922.

Data of demographic and laboratory findings of all infants (male and female) with diagnosed SW CAH are presented in Table 3. Comparing both genders together and only male infants, the age at diagnosis (days), mean serum sodium and potassium and frequency of severe hyponatremia was similar at presentation between the screened and unscreened groups. However, weight loss (%) from birth to the day of diagnosis was greater in the unscreened cohort (−6.2 ± 6.9 vs. −0.3 ± 4.9, *p* = 0.048).

Time (median) from birth to the day of SW CAH diagnosis in the unscreened cohort was 15.7 (0–42) days, screened cohort—13.25 (1–32) days, *p* = 0.78. There were significant differences in weight SDS at the diagnosis between unscreened and screened newborns in the whole cohort (−0.065 ± 1.36 vs. 0.1 ± 0.87 SDS, respectively, *p* = 0.049). In addition, significant differences in weight at diagnosis between male newborn groups were observed (−1.67 ± 1.12 vs. 0.046 ± 1.01 SDS in unscreened and screened boys, respectively, *p* = 0.001).

There was no significant correlation between weight and age (in days) when CAH was confirmed, *r* = −0.44, *p* = 0.782. Importantly, unscreened (males and females) infants lost 6.2 ± 6.9% of weight from their birth to the day when CAH was confirmed and screened infants with SW CAH gained 6.7 ± 1.3% of weight, *p* = 0.007 (Figure 2). The difference in weight change between the male newborns’ groups was 13.2%. Eight out of 25 unscreened patients (32%) and 1 out of 9 (11.1%) screened patients were treated in the neonatal intensive care unit (*p* = 0.22).

On the day of diagnosis, serum potassium levels in unscreened all cohort and separately in males’ group were higher, although not statistical significantly, than in screened cohort (6.9 ± 1.6 versus 6.7 ± 1.3, *p* = 0.79 and 7.7 ± 1.5 mmol/L versus 6.89 ± 1.5 mmol/L, *p* = 0.33 respectively). Hyperkalemia (>6.9 mmol/L) occurred in 63.6% of patients in the unscreened males’ cohort and in 55.5% of screened males, *p* = 0.65. No significant correlation was observed between serum potassium levels and days when CAH was confirmed (*r* = 0.202, *p* = 0.195). Serum sodium levels were 124.5 ± 9.7 mmol/L (102–138) in the unscreened males and 126.31 ± 8.99 mmol/L (111–135.8) in screened males, *p* = 0.64; hyponatremia (<130 mmol/L) was determined in 72% and 55% of unscreened and screened newborn’s males, respectively, *p* = 0.59.

In both groups of male infants combined (unscreened and screened), clinical symptoms at the diagnosis were: vomiting in 21 (67.7%), jaundice in 9 (29%), hypotonia in 11 (35.5%) and hypovolemic shock in two (6.5%) patients. Three patients (9.7%) had hypoglycemia and 22 (70.9%)-hyperkalemia. Clinical signs and symptoms distribution in males’ groups (unscreened and screened) are shown in Figure 3.

### 3.3. Phenotype and Genotype Correlation in Patients of CAH Diagnosis in Lithuania Cohort

Over a 6-year period (2015–2020), all screened patients were genotyped as part of the clinical follow-up and 12 patients had SW (9 males and 3 females) form and one female of CAH.

Six patients with blood levels of 17OHP less than 300 nmol/L (30.8–180) during NBS, were re-examined with ACTH (Synacten) stimulation test, and SW CAH form was confirmed for five patients and for one girl was confirmed SV CAH at the 71 days of age. Two cases were twin brothers with 17-OHP levels above 30 nmol/L (46.3 and 32.2 nmol/L) during NBS, but no clinical signs were observed. During ACTH stimulation tests maximal stimulated 17OHP levels were 31.0 and 28.2 nmol/L and genetic analysis confirmed pathogenic mutation (c.293-13C > G) only in one allele of *CYP21A2* gene. One girl without clinical manifestation of CAH had 17OHP levels above the normal range (39.8 nmol/L) on NBS, and a benign compound heterozygous mutation was identified. In another girl with mild clitoriomegaly (Ist virilization stage according to Prader scale) with and basal 17OHP level of 697 nmol/L during NBS and stimulated 17OHP of 580 nmol/L, molecular analysis of *CYP21A2* gene revealed a pathological mutation in only one allele.

Molecular genetic analysis was available for 52 unrelated individuals in the study group (Table 4). The diversity of the identified pathogenic variants, allele frequencies, and expected phenotype are shown in Table 5. Large deletions and conversions extending to about 30 kb or covering several exons of *CYP21A2* genes comprised 41% of CAH alleles in our group. Three novel pathogenic variants in *CYP21A2* gene were identified.

The most common pathogenic variants of the *CYP21A2* gene were 30 kb deletion and c.293-13C > G, accounting for 21% and 19% of all pathogenic variants, respectively. Large deletions involving several exons of *CYP21A2* gene accounted for as much as 41% of all pathogenic variants of *CYP21A2* gene. Two novel pathogenic variants c.329T>A (p.(Leu110Ter) and c.525C > A(p.(Tyr175Ter)) not derived from the pseudogene and both generating a premature termination codon were identified in the compound heterozygous state for a male patient with salt-wasting CAH. Novel frameshift variant c.916del (p.(Val306PhefsTer17)) was identified in the compound heterozygous state with pathogenic variant c.518T>A (p.(Ile173Asn)) for two brothers both manifesting with SW CAH.

## 4. Discussion

CAH belongs to the group of rare diseases and its prevalence in Lithuania is similar to other countries of the world. Annelieke A A van der Linde and co-authors indicated a worldwide prevalence of CAH to be approximately 1:10,000 to 1:20,000 [13]. It was considered that the SW CAH form accounts for approximately 75% of classical CAH cases. During the 6 years of NBS in Lithuania, an unusually high rate of SW CAH was detected (*n* = 12; 92%) as compared to only one case of SV CAH (8%) was detected. The different distribution of SW and SV incidence rates was described in other population-based studies, namely in Japan 73:14 and in New York State 90:8, respectively [14,15]. A relatively short period of NBS could be the reason for such a different prevalence of the two CAH forms. Alternatively, the introduction of NBS most probably increases the rates of detection of CAH patients that would have been partly overlooked in regular clinical practice, and the ratio of SW and SV forms might change in the future.

The NBS program is performed in many developed countries, however, there are only a few published studies that evaluated the NBS program effectiveness. In a study by Heather et al. in New Zealand, the time of diagnosis of CAH was compared before and after the introduction of NBS (11 ± 3.0 (0–26) days versus 8.5 ± 3.2 (3–16) days, respectively, *p* = 0.003) [2]. In another study that assessed the effectiveness of NBS in Canada, when only male infants were included, the time at diagnosis averaged 14 days (1–30) prior to NBS program and 5.5 days (0–13) after the introduction of NBS program (*p* = 0.077) [16]. Considering that blood samples for NBS were collected no earlier than 48 h after birth, and the Canadian study did not provide precise information as to why the time to diagnosis ranged from 0 days, it might be assumed that in some cases, CAH was diagnosed prenatally. In our study, delay in shipment and delivery of DBS to NBS laboratory (DBS samples were received on 3–19 days of life) might lead to a lack of significant difference between the time of diagnosis in screened and unscreened newborns. Although in our study the time to diagnosis was not significantly different in the screened and unscreened groups, the 10-day difference in neonates was clearly clinically relevant in the case of the salt-wasting form of CAH. The timeline of the NBS is becoming earlier worldwide because of increasing numbers of inborn errors of metabolism added to the screening panel, requiring immediate intervention [17], and in the U.S. the recommended age in days when the first results were obtained should be seven days of life [18]. The median age at the time of positive screening results differs among countries—from 6 days in Japan [16] to 8.7 days in Sweden [19], 13 days in Texas [20] and 17 days in Brazil [21]. In Lithuania, the main reason for the delay of diagnosis was the delay in delivery of the NBS cards to the laboratory, clearly having significant implications and should be targeted to improve the clinical usefulness and effectiveness of the screening.

Adrenal crisis is a life-threatening medical emergency, and 55% of screened and as much as 72% of unscreened male newborns had hyponatremia <130 mmol/L at the day of diagnosis. Severe adrenal crisis during the neonatal period may cause permanent neurological comorbidities. Studies in Japan before the NBS and in the UK showed that up to 20% of patients with the SW form of 21OHD developed learning difficulties and intellectual disabilities [16,22,23]. The risk of developing severe salt-wasting increases during the second week of life, and, therefore, an early diagnosis and intervention, preferably during the first week after birth, is of great importance [24,25].

The possibility of developing severe salt-wasting is extremely low without weight loss, which in our cohort was significantly greater in the unscreened group of newborns, even if there were no significant differences in sodium levels at the diagnosis in screened and unscreened newborns. Change in weight from birth to the day of diagnosis is an important indicator of the newborn’s health status. Physiological weight loss in neonates is found to be up to 10% of birthweight [26] and weight regain usually occurs around 9–10 days postpartum [27]. Clinical symptoms of adrenal insufficiency, such as excessive regurgitation and vomiting, also occurred also significantly more frequently in the unscreened cohort, despite a similar biochemical profile in both groups. Of note, there were no cases of hypotension and hypovolemic shock in the group of screened patients, compared to 44% and 8% of unscreened newborns, respectively. Similarly, in the aforementioned Canadian study, the biochemical profile of the screened and unscreened newborns was comparable, but their clinical status seemed to differ, based on a lesser need for medical transport and fluid bolus, and shorter hospital length of stay in screened infants, there were no salt-wasting crises in the screened cohort (as defined by a need for resuscitation), compared with 25% of subjects in the unscreened cohort [16,28]. In our study, one screened newborn had SW crisis and was admitted to NICU with consider-able delay on day 14 due to parental neglect despite the timely information on the disease of their child. Similar data on delayed arrival at the hospital were also published by other authors [29].

The sensitivity of NBS program for CAH in Lithuania in 2015–2020 was 100% as no false-negative cases were found and the specificity—>99.9%, which were comparable to previously published results from other countries [30]. However, because of the high false-positive tests rate, the positive predictive value of the NBS was only 4%, which was also comparable to other countries, using immunoassay techniques for measurement of 17OHP levels [31]. Although stratifying the cut-off levels by gestational age have been shown to improve the positive predictive value to some extent, its efficiency is still limited, since false positive results include the natural increase of 17OHP due to stress stimuli, critical illness, adrenal immaturity as well as cross-reactivity of the immunoassay antibody with other hormones and endogenous compounds in blood [17,32,33]. Significant reduction of the false-positive results and further improvement of the NBS may be achieved when a second-tier test is used [16]. Alternatively, liquid chromatography-tandem mass spectrometry (LC-MS/MS) is considered as the gold standard for steroids assays; therefore, since 2018 the Endocrine Society guidelines recommend using the LC-MS/MS for the measurement of 17OHP of the second tier in neonatal screening [9].

Interestingly, we documented elevated basal and stimulated 17OHP concentrations in three carriers of a *CYP21A2* gene mutation in a single allele or benign compound mutation in *CYP21A2.* Italian study describes three girls with slightly elevated 17OHP levels (18.2–35.3 nmol/L) detected during NBS and transient clitoromegaly were later identified as heterozygous cases (for all cases were detected c.293-13C > G pathogenic variant in one allele of *CYP21A2* gene). In these cases, spontaneous regression of clitoromegaly was observed during the follow-up through the first six months of life [34]. In other studies, higher levels of mean stimulated 17OHP in the carriers of *CYP21A2* mutation presented with premature pubarche and late-onset hyperandrogenemia [35,36,37].

We did not evaluate the cost-efficacy of NBS, and the data from the literature are rather controversial. Earlier studies reported CAH screening would be either unlikely to be cost-effective or cost-effective only if a single test is performed [19,38]. While recent studies in Canada, Brazil and Sweden concluded that NBS screening for CAH enables to reduce morbidity and mortality [39,40] as well as costs associated with adverse outcomes, transportation, and hospitalization length [41]. The long-term health benefits of NBS for CAH are unknown and should be addressed in future studies.

The main strength of this study is that we analyzed all cases of CAH after NBS implementation and almost whole unscreened patients’ cohort in Lithuania. On the other hand, the main drawback of this study is the low number of CAH patients confirmed by NBS, which limits the statistical power of data analyses.

## 5. Conclusions

The introduction of the NBS program for CAH in Lithuania resulted in a more favorable clinical presentation of affected male infants compared to those diagnosed before NBS, despite the non-significant difference in time of diagnosis, which should definitely be shortened by timely DBS shipment from maternity departments to NBS laboratory. The long–term effects of NBS and the consequences of hyponatremia in the neonatal period on neurological development, intellectual functioning, and quality of life should be explored in future studies. Discrepancies in genotype and biochemical and clinical profiles described in our patients confirm earlier published data in CAH patients.

## Figures and Tables

**Figure 1 medicina-57-01035-f001:**
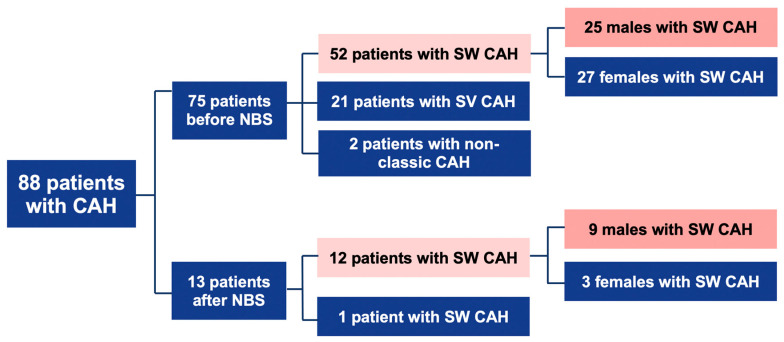
Descriptive scheme of cases. CAH: congenital adrenal hyperplasia; NBS: newborn screening; SW: salt-wasting; SV: simple virilizing.

**Figure 2 medicina-57-01035-f002:**
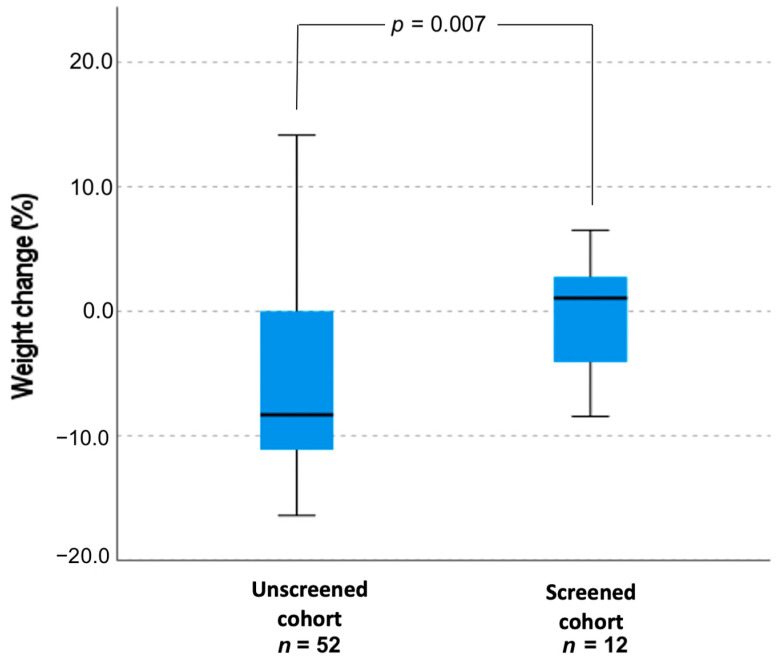
Weight change (%) in unscreened and screened patients’ cohort with SW CAH.

**Figure 3 medicina-57-01035-f003:**
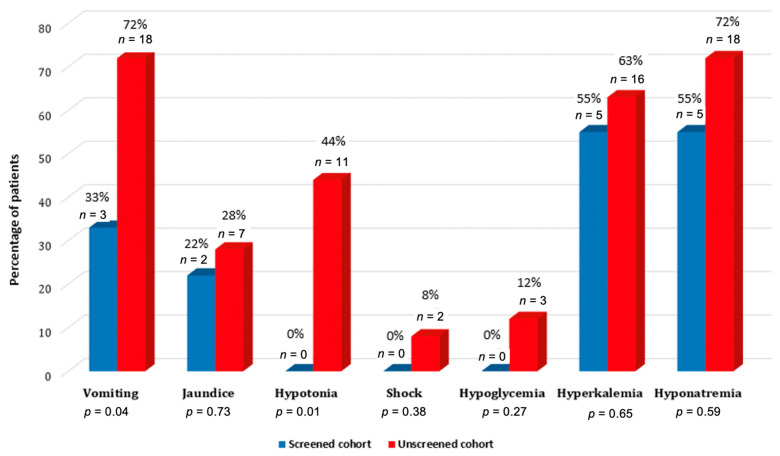
The clinical signs and symptoms in unscreened and screened males at the day of diagnosis.

**Table 1 medicina-57-01035-t001:** Cut-off values of 17-hydroxyprogesterone (17OHP) in DBS.

Gestational Age (Weeks)	Cut-Off of 17OHP Serum Concentrations (ng/mL Serum *)	Reference Value to Recall for a Second Sample (ng/mL Serum *)	Reference Value to Urgent Medical Evaluation (ng/mL Serum *)
>38	28	>28 <33.6	≥33.6
35–37	39	>39 <46.8	≥46.8
32–34	53	>53 <63.6	≥63.6
29–31	80	>80 <96	≥96
<29	198	>198 <237.6	≥237.6

* Conversion factor: 1 ng/mL serum = 1/0.66 nmol/L blood. DBS: dried blood spot.

**Table 2 medicina-57-01035-t002:** Results of CAH screening from 2015 to 2020.

	Numbers of Newborns
Screened newborns (2015–2020)	158,486
Positive tests (>cut off 17OHP): Recalled for the 2nd sample Notifications of positive case: Referrals for urgent medical evaluation after 1st DBS Referrals for medical evaluation after 2nd DBS	320 204 118 116 2
False-positive tests	307
True positive tests: From recalled for the 2nd sample From referred for urgent medical evaluation	13 2 11
False-negative tests *	0

* No notifications of false-negative cases were recorded in the period from January 2015 to July 2021; CAH: congenital adrenal hyperplasia.

**Table 3 medicina-57-01035-t003:** Age at diagnosis, weight change, and biochemical profile is screened and unscreened infants with SW CAH.

Variables	Screened Cohort (*n* = 12)	Unscreened Cohort (*n* = 52)	Screened Males (*n* = 9)	Unscreened Males (*n* = 25)
Age at diagnosis in days, mean ± SD	14.6 ± 9.6	16.5 ± 11.6	15.44 ± 7.79	19.13 ± 7.156
	*p* = 0.603		*p* = 0.19	
Weight change (%), mean ± SD	−0.31 ± 4.9	−6.2 ± 6.9	5.42 ± 17.72	−7.81 ± 8.58
	*p* = 0.007		*p* = 0.023	
Potassium mmol/L, mean ± SD	6.7 ± 1.3	6.9 ± 1.6	6.89 ± 1.5	7.7 ± 1.5
	*p* = 0.79		*p* = 0.33	
Sodium, mmol/L, mean ± SD	127.5 ± 7.9	127.6 ± 11.6	126.31 ± 8.99	124.5 ± 9.7
	*p* = 0.67		*p* = 0.64	
Sodium < 132 mmol/L, n (%)	9 (75%)	29 (56.4%)	7 (78%)	20 (80%)
	*p* = 0.22		*p* = 0.89	
Sodium < 130 mmol/L, n (%)	5 (42%)	25 (48%)	5 (55%)	18 (72%)
	*p* = 0.67		*p* = 0.59	

SW: salt-wasting.

**Table 4 medicina-57-01035-t004:** *CYP21A2* disease-causing variants detected in the Lithuanian patients (1989–2020).

Pahtogenic Variant		Identification No. (dbSNP [10], HGMD [11])	Expected Phenotype	Allele No. (Frequency) (*n* = 94)
cDNA Level (NM_000500.9, NG_007941.3)	Predicted Protein Change (NP_000491.4)			
c.1A > C	p.(Met1Leu)		SW	1(0.01)
c.293-13C > G	New splice acceptor site	rs6467	SW	16 (0.17)
c.[293-13C > G;1360C > T]	p.[(?);(Pro454Ser)]	-	SW	3 (0.03)
c.[293-13C > G;332_339delGAGACTAC]	p.[(?);(Gly111Valfs)		SW	1 (0.01)
c.329T > A	p.(Leu110Ter)	-	SW	2 (0.02)
c.332_339delGAGACTAC	p.(Gly111Valfs)			1 (0.01)
c.518T > A	p.(Ile173Asn)	rs6475	SV	9 (0.10)
c.525C > A	p.(Tyr175Ter)	-	SW	1 (0.01)
c.844G>	p.(Val282Leu)	rs6471	NC	4 (0.04)
c.916del	p.(Val306PhefsTer17)	-	-	1 (0.01)
c.923dupT	p.(Leu308Phefs)	rs267606756	SW	1 (0.01)
c.[923dupT;955C > T]	p.[(Leu308Phefs); (p.Gln319Ter)]	-	SW	1 (0.01)
c.955C > T	p.(Gln319Ter)	rs7755898	SW	3 (0.03)
c.1069C >	p.(Arg357Trp)	rs7769409	SW	4 (0.04)
c.1294G > A	p.(Glu432Lys)	rs1245238711	NC	1 (0.01)
c.1360C > T	p.(Pro454Ser)	rs6445	NC	2 (0.02)
~30 kb deletion ^1^	p.(0)	-	SW	20 (0.21)
Deletion of exons 1–3	p.(0)	-	SW	7 (0.07)
Deletion of exons 1–7	p.(0)	-	SW	10 (0.11)
Unidentified pathogenic variant		-		6 (0.06)

^1^ ~30 kb deletion causing the formation of chimeric gene CYP21A1P(NR_040090.1)n.877+?_CYP21A2(NM_000500.9)c.477+?del. is suspected from MLPA results, but Sanger sequencing is required to confirm the genotype; SW: salt-wasting; SV: simple virilizing; NC: non-classical.

**Table 5 medicina-57-01035-t005:** Genotypes of all investigated patients due to 21OHD in Lithuania.

Genotype ^1^	Age at Diagnosis	Clinical Manifes-Tation	Expected Phenotype	Gender
1989–2014				
c.[CYP21A1P(NR_040090.1)n.877+?_CYP21A2 (NM_000500.9)c.477+?del)];[CYP21A1P(NR_040090.1)n.877+?_CYP21A2(NM_000500.9)c.477+?del)], p.[(0)];[(0)] ^2^	1–5 days	SW	SW	2 females
c.[CYP21A1P(NR_040090.1)n.877+?_CYP21A2(NM_000500.9)c.477+?del)];[(?-8)_(447+1_448-1)del], p.[(0)];[(0)] ^3^	6–20 days	SW	SW	2 females, 3 males
c.[CYP21A1P(NR_040090.1)n.877+?_CYP21A2(NM_000500.9)c.477+?del)];[(?-8)_(939+1_940-1)del], p.[(0)];[(0)] ^4^	0 day	SW	SW	1 female
c.[923dupT];[1451G>C], p.[(Leu308PhefsTer6(;)Arg484Pro)]	17 days	SW	SV	1 male
c.[CYP21A1P(NR_040090.1)n.877+?_CYP21A2(NM_000500.9)c.477+?del)];[(?-8)_(939+1_940-1)del], p.[(0)];[(0)] ^4^	13 days	SW	SW	1 male
c.[CYP21A1P(NR_040090.1)n.877+?_CYP21A2(NM_000500.9)c.477+?del)];[(?-8)_(939+1_940-1)del], p.[(0)];[(0)] ^4^	34 days	SW	SW	1 male
c.[CYP21A1P(NR_040090.1)n.877+?_CYP21A2(NM_000500.9)c.477+?del)];[293-13C>G], p.[(0)];[(?)]	0–17 days	SW	SW	1 female, 2 males
c.[(?-8)_(549+1_550-1)del];[293-13C>G], p.[(0)];[(?)]	13 days	SW	SW	1 male
c.[(?-8)_(939+1_940-1)del];[293-13C>G;518T>A;1360C>T], p.[(0)];[(?;Ile173Asn;Pro454Ser)]	15 days	SW	SW	1 male
c.[CYP21A1P(NR_040090.1)n.877+?_CYP21A2(NM_000500.9)c.477+?del)];[955C>T], p.[(0)];[(Gln319Ter)]	42 days	SW	SW	1 female
c.[CYP21A1P(NR_040090.1)n.877+?_CYP21A2(NM_000500.9)c.477+?del)];[1069C>T], p.[(0)];[(Arg357Trp)]	30 days	SW	SW	1 male
c.[(?-8)_(447+1_448-1)del];[293-13C>G], p.[(0)];[(?)]	26 days	SW	SW	1 male
c.[(?-8)_(447+1_448-1)del];[923dupT], p.[(0)];[(Leu308PhefsTer6)]	27 days	SW	SW	1 male
c.[(?-8)_(447+1_448-1)del];[923dupT; 955C>T], p.[(0)];[(Leu308PhefsTer6;Gln319Ter)]	12 days	SW	SW	1 female
c.[1A>C];[293-13C>G], p.[(Met1Leu)];[(?)]	22 days	SW	SW	1 male
c.[293-13C>G];[293-13C>G;1360C>T], p.[(?)];[(?;Pro454Ser)]	1 day	SW	SW	1 female
c.[293-13C>G];[c.332_339del], p.[(?)];[(Gly111ValfsTer21)]	21 days	SW	SW	1 male
c.293-13C>G(;)332_339del(;)518T>A, p.(?)(;)(Gly111ValfsTer21)(;)(Ile173Asn)	6 days	SW	SW/SV	1 male
c.293-13C>G(;)332_339del(;)1360C>T, p.(?)(;)(Gly111ValfsTer21)(;)(Pro454Ser)	15 days	SW	SW/NC	1 female
c.[293-13C>G];[518T>A], p.[(?)];[(Ile173Asn)]	11 days–12.9 years	SV	SV	3 females, 1 male
c.293-13C>G(;)955C>T(;)1069C>T, p.(?)(;)(Gln319Ter)(;)(Arg357Trp)	30 days	SW	SW	1 male
c.[293-13C>G];[1069C>T], p.[(?)];[(Arg357Trp)	7 days	SW	SW	1 female
c.844G>T(;)955C>T, p.(Val282Leu)(;)(Gln319Ter)	6,2 years	SW	NC	1 female
c.955C>T(;)1360C>T(;)1455del, p.(Gln319Ter)(;)(Pro454Ser)(;)(Met486TrpfsTer56)	3 days	SW	SW/NC	1 female
c.[518T>A];[518=], p.[(Ile173Asn)];[(Ile173=)]	6.3 years	SW	SV?	1 female
c.[(?-8)_(939+1_940-1)del];[518T>A], p.[(0)];[(Ile173Asn)]	5 years	SV	SV	1 female
c(?-8)_(939+1_940-1)del];[518T>A], p.[(0)];[(Ile173Asn)]	9 years	SV	SV	1 male
c.[(?-8)_(939+1_940-1)del];[844G>T], p.[(0)];[(Val282Leu)]	7 days	SV	NC	1 female
c.[(?-8)_(939+1_940-1)del]];[1294G>A], p.[(0)];[(Glu432Lys)]	5 years	SV	NC	1 female
c.[293-13C>G;1360C>T];[1360C>T], p.[(?;Pro454Ser)];[(Pro454Ser)]	22.7 years	NC	NC	1 female
c.293-13C>G(;)518T>A(;)1360C>T, p.(?)(;)(Ile173Asn)(;)(Pro454Ser)	27 years	SV	SV/NC	1 male
Heterozygous duplication of *CYP21A2* gene. Homozygous or heterozygous genotype for c.955C>T, p.(Gln319Ter) variant	5 years	SV	SW	1 male
c.[CYP21A1P(NR_040090.1)n.877+?_CYP21A2(NM_000500.9)c.477+?del)];[844G>T]	16 years	NC	NC	1 female
c.[844G>T];[=], p.[(Val282Leu)];[(Val282=)]	56 days	NC	-	1 female
2015–2020				
c.[CYP21A1P(NR_040090.1)n.877+?_CYP21A2(NM_000500.9)c.477+?del)];[CYP21A1P(NR_040090.1)n.877+?_CYP21A2(NM_000500.9)c.477+?del)], p.[(0)];[(0)]	8 days	SW	SW	1 male
c.CYP21A1P(NR_040090.1)n.877+?_CYP21A2(NM_000500.9)c.477+?del)(;)(?-8)_(939+1_940-1)del], p.[(0)];[(0)]	8 days	SW	SW	1 male
c.CYP21A1P(NR_040090.1)n.877+?_CYP21A2(NM_000500.9)c.477+?del)(;)(c.293-13C>G), p.[(0)];[(?)]	8 days	SW	SW	1 female
c.(?-8)_(939+1_940-1)del(;)293-13C>G, p.[(0)];[(?)]	10 days	SW	SW	1 male
c.(?-8)_(939+1_940-1)del(;)329T>A, p.[(0)];[(Leu110Ter)]	20 days	SW	SW	1 male
c.293-13C>G(;)(293-13C>G)	5 days	SW	SW	1 female
c.[293-13C>G];[518T>A], p.[(?)];[(Ile173Asn)]	32 days	SW	SV	1 male
c.[293-13C>G];[1069C>T], p.[(?)];[(Arg357Trp)	15 days	SW	SW	1 male
c.329T>A(;)525C>A, p.(Leu110Ter)(;)(Tyr175Ter)	14 days	SW	SW	1 male
c.[518T>A];[916del], p.[(Ile173Asn)];[(Val306PhefsTer17)]	43 days	SW	SV	1 male
c.[518T>A];[916del], p.[(Ile173Asn)];[(Val306PhefsTer17)]	15 days	SW	SV	1 male
c.955C>T(;)1069C>T p.(Gln319Ter)(;)(Arg357Trp)	2 days	SW	SW	1 female
c.[518T>A];[518=], p.[(Ile173Asn)];[(Ile173=)]	71 days	SV		1 female

^1^ According to HGVS Recommendations for the Description of Sequence Variants [12]. Reference sequences: NM_000500.9; NP_000491.4; NG_007941.3. ^2^ Homozygous genotype for ~30 kb deletion causing the formation of chimeric gene CYP21A1P(NR_040090.1)n.877+?_CYP21A2(NM_000500.9)c.477+?del and pathogenic variant c.293-13C > G is suspected from MLPA results. ^3^ Compound heterozygous genotype for ~30 kb deletion causing the formation of chimeric gene and for deletion of exons 1–3. ^4^ Compound heterozygous genotype for ~30 kb deletion causing the formation of chimeric gene and for deletion of exons 1–7; SW: salt-wasting; SV: simple virilizing; NC: non-classical.

## Data Availability

We did not report any data.

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
