# Peer review of "Impact of Newborn Screening on Clinical Presentation of Congenital Adrenal Hyperplasia"

_medicina, 2021, doi:10.3390/medicina57101035_

Round 1
Reviewer 1 Report
The authors present an evaluation of the impact of newborn screening on the clinical presentation of CAH.
There are a number of minor corrections required some examples are:
Program or programme – both are used
Line 86 currier
Line 148: life births
Line 258: salt waisting CAH
Line 326: some extend should e extent
There are also a number of observations or queries for which responses may strengthen the paper:
Line 51: During the prenatal period, testosterone hypersecretion does not result in any visible signs for male infants and adrenal insufficiency occurs at 2 to 3 weeks of age with severe hyperkalaemia, hyponatraemia, and hypovolemic shock [6].
The authors then correctly go on to indicate the classifications of CAH indicating not all males develop adrenal insufficiency. Perhaps that sentence is not required
Lines 61-66: In order to prevent life-threatening adrenal crisis and to help make the appropriate sex
assignments in affected female patients, newborn screening (NBS) programs for CAH have been introduced in many countries. Measurements of 17-hydroxyprogesterone (17OHP) levels have been used as a marker for the 21OH deficiency. Neonatal screening for the CAH provides the opportunity for early detection and treatment and has been implemented in many countries [7].
The first and last sentences are essentially the same
Line 87: The level of 17-hydroxyprogesterone (17OHP) in the DBS was measured by fluorometric enzyme immunoassay (FEIA). Please provide further details of the method
Table 1: The authors use dried blood spot samples yet report results in ng/mL of serum.
What conversion is used to modify results obtained in blood to serum
In 2.2 there is detail of a positive case for salt-wasting CAH. However, the authors indicate inclusion of non-classical forms of CAH at least in the pre-screening period. Is non-classical CAH being actively sought during the screening phase? Please provide a definition of cases.
Further – if non classical is sought how is it confirmed that there are no missed cases
Figure 1: During screening 12 cases of salt wasting CAH were detected 9 males and 3 females with gender as indicated on the DBS card. Is there any comment on the sex ratio disparity or whether those indicated as male actually were?
Line 157: 3 with ambiguous genitalia
Line 164: Three patients were suspected of having CAH on the first day of life because of ambiguous genitalia (n=2) and one was the sibling of a previously confirmed CAH patient.
Were there 2 or 3 cases with ambiguous genitalia?
Table 1 and Line 155: In total, 118 newborns with abnormal 17-OHP levels on screening were referred for medical evaluation. How many newborns were recalled for a second sample as opposed to urgent medical evaluation?
Line 187: There were significant differences between the groups (unscreened all cohort vs. screened and unscreened males and screened males) in weight SDS at the diagnosis: -0.065±1.36 SDS versus 0.1±0.87 SDS, p=0.049 and -1.67±1.12 SDS versus 0.046±1.01 SDS, in unscreened and screened boys, respectively, p=0.001.
Redundancy with males being included in the bracket and then again in the text
Line 191: There was no significant correlation between weight on the day of diagnosis and the day when CAH was confirmed, r= -0.44, p=0.782. Please define the time difference between “day of diagnosis” and time taken for “day confirmed”.
Figure 3: If only half the males had hyperkalemia and/or hyponatremia at diagnosis – how were the cases defined as salt-wasting
Line 224: 30 nmol/L (46.3 and 32.2 ng/mL). The authors use different units of ng/mL or nmol/L on a number of occasions throughout the tables and text. A prime example is detailed here
Line 285: In Lithuania, the main reason in delay of diagnosis is delay in delivery of the NBS cards to the laboratory, clearly having significant implications and should be targeted to improve the clinical usefulness and effectiveness of the screening.
In the results section; line 161: On average, the screening DBS samples of confirmed CAH cases were collected on day 3 (median) (range: 2-4) and reached the screening laboratory by mail or courier on day 9 (range 3-19), and the 17OHP result was available on day 13 (range: 7-23 days).
There is no discussion of the time taken between receipt of sample and result availability which is well within the control of the laboratory
Line 289: Adrenal crisis is a life-threatening medical emergency, and the majority of the unscreened and screened newborns had hyponatraemia < 130 mmol/l at the day of diagnosis.
The results in Table 3 indicate that this is correct for less than half of the cases.
Line 316: As a result of our data (in table 5), the NBS is effective for SV CAH diagnosis and SV.
CAH form was detected clinically significantly earlier in a screened group comparing with unscreened (71 days versus 27 years), and it is a very important aspect for outcomes of final adult height, expensive treatment for precocious puberty.
With only 1 case of SV detected in the screened group it is not proof of effectiveness of NBS for SV.
References 2 and 13 are the same article. Please correct
Author Response
Dear Reviewer,
Thank You for your very accurate comments and questions.
Our response to the Your comments questions is attached as a Word document.
Please see the attachment.
Kind regards,
Ruta Navardauskaite

Reviewer 2 Report
The mauscript by Navardauskaitė et al is interesting. I have the following comments:
- Abstract: Only 1 case of SV but 12 cases of SW were found by NBS. You wonder if some SV were missed by NBS…
- Introduction, first paragraph: Recently it has been claimed 95-99% of CAH is 21OHD and a more modern extensive CAH review could be used (e.g. Claahsen-van der Grinten HL et al Endocr Rev. 2021) instead of e.g. ref 1.
- Table 1: 17OHP is given in ng/mL while in the text below in nmol/L.
- Line 122: Please give the rational why only males with SW were analyzed.
- Line 133: t-test? Mann Witney U test? Please describe in more details.
- Table 2: How can you know there were no false negatives? It may take years before an undiagnosed SV case is diagnosed clinically.
- Results: The large draw back of this study is the low number of screened individuals and hence the number of included CAH patients. This will make many results insignificant. This should be discussed in the Limitations.
- Line 162-164; There is quite a delay from the sample taken until the result. Did anyone go into SW crisis during this delay?
- Line 289: Maybe add a reference to a pediatric adrenal crisis review, e.g., Rushworth RL et al Horm Res Paediatr 2018.
- Line 318: Were really all SV CAH diagnosed at the age of 27 years before screening?
- Line 349-350: NBS seems to improve long term clinical outcomes in CAH and this was recently reviewed (Lajic S Int J Neonatal Screen. 2020). This should be added.
Author Response

(The authors gave the same response as above.)

Round 2
Reviewer 1 Report
Thank you for resubmitting this article. It is much improved over the original version.
However, there remain some queries that need to be addressed:
Please note in newborn screening a positive test is any result where the action requires further sampling from the baby. It is confusing exactly how many babies had a positive screen
Table 2: Needs further detail to include the patients where a repeat sample was required
In total 322 infants required further testing, 204 required a repeat DBS and 118 required a medical evaluation. The positive predictive value is therefore 0.04. - Is this correct? Please indicate if any of the infants requiring repeat DBS had CAH
Line 248: above the normal range (39.8 ng/mL) - Are the units ng/mL or nmol/L?
In a study by 286 Heather et al. in New Zealand, the time of diagnosis of CAH was compared before and 287 after the introduction of NBS (11±3.0 (0-26) days versus 8.5±3.2 (3-16) days, respectively, 288 p = 0.003) [2]. Please note this is not reference 2. Apparently the correction of duplication of reference 2 and 13 in the original has not been checked in detail. All reference numbers in the "Discussion" require review
Line 334: In our study, one screened newborn had SW crisis and in NICU.... Requires correction of the english
line 345 and line 11: Check the PPV after inclusion of repeat dbs samples
Author Response
Dear Reviewer,
Thank you for your very accurate remarks and insights, which to help to improve our manuscript value.
We corrected our manuscript according to your comments.
Kind regards,
Ruta Navardauskaite

Reviewer 2 Report
The revised mauscript by Navardauskaitė et al has improved greatly. I have only 1 minor comment:
- Please comment the high ratio SW/SV in the Discussion.
Author Response
Dear reviewer,
Thank you for your very accurate remarks and insights, which to help to improve our manuscript value.
We filled the discussion part with an answer to your question about the high ratio of SW/SV:
It is considered that SW CAH form accounts for approximately 75% of classical CAH cases. During the 6 years of NBS in Lithuania, unusually high rate of SW CAH (n = 12, 92%) as compared to only 1 case of SV CAH (8%) was detected. The different distribution of SW and SV incidence rates is described in other population-based studies, in Japan 73:14, in New York State 90:8, respectively [14,15]. A relatively short period of NBS could be the reason of such different prevalence of the two CAH forms. Alternatively, the introduction of NBS most probably increases the rates of detection of CAH patients that would have been partly overlooked in regular clinical practice, and the ratio of SW and SV forms might change in the future.
